# Development of a Low-Cost Wireless Phonocardiograph With a Bluetooth Headset under Resource-Limited Conditions

**DOI:** 10.3390/medsci6040117

**Published:** 2018-12-17

**Authors:** Himel Mondal, Shaikat Mondal, Koushik Saha

**Affiliations:** 1Department of Physiology, Fakir Mohan Medical College and Hospital, Balasore, 756019, Odisha India; 2Department of Physiology, Kalna SD Hospital, Kalna, Purba Bardhaman, 713409, West Bengal, India; drshaikat@gmail.com; 3Department of Anatomy, Fakir Mohan Medical College and Hospital, Balasore, 756019, Odisha, India; jkoukou@gmail.com

**Keywords:** auscultation, cell phone, heart sounds, phonocardiography, stethoscopes

## Abstract

Commercially available digital stethoscopes can be used as a phonocardiograph to record heart sounds. However, procuring a costly digital stethoscope may not be possible under resource-limited conditions. A low-cost, wire connected, and mobile phone-assisted phonocardiograph has been reported previously. The aim of this study was to develop a low-cost and wireless phonocardiograph for resource-limited settings. A Bluetooth headset was dismantled to find its microphone. A stethoscope ear knob was cut to make a small bell and it was attached to the microphone. This modified Bluetooth headset was capable of recording sounds while connected to a mobile device with audio recording application. The modified Bluetooth headset, mobile phone, and audio recording software can serve as a wireless phonocardiograph (WiPCGh). Heart sounds were successfully recorded with the help of the newly developed WiPCGh. The audio files were shared with a personal computer (PC) via Bluetooth. The wave form was analyzed in a PC-based audio editing application. First and second heart sounds with systolic and diastolic murmur were identified. WiPCGh can be utilized in recording heart sounds for academic and telemedicine purposes. However, the capability of WiPCGh in the diagnosis of cardiac diseases is yet to be explored in future studies.

## 1. Introduction

A 20-year-old patient, suffering from ventricular septal defect recorded heart sounds by simply holding his mobile phone on the chest wall. When the physician heard the sound and analyzed it using audio editing software, the result was impressive [1]. Later, Bhaskar developed a simple digital stethoscope with the help of a stethoscope chestpiece and an earphone. The author suggested that this earphone may be either connected to a mobile device or a personal computer to record the heart sounds [2]. Mamorita et al. used a similar type of device while developing an application for phonocardiography [3]. This type of earphone-based device was also reported by Bhimani et al. [4]. In addition, Thomas et al. described recording of heart sounds by 3 other types of phonocardiographs based on mobile devices [5]. A pictorial comparison of available mobile phone-assisted phonocardiographs is shown in Figure 1.

Costly digital stethoscopes have been on the market for a long time. However, the devices shown in Figure 1 can be built in any resource-limited settings with minimal instruments. The cost of the device is also very low. These devices can help in recording heart sounds, especially for academic purposes. Recorded audio files can be used as a teaching aid for medical students. As these devices are connected either to tubing or to a wire, the devices require the presence of a mobile phone near, the subject and this may alter the normal heart rate [6]. It may also alter the heart rate variability [7]. In contrast, Tamer et al. reported that no hemodynamic disturbance occurs due to presence of a mobile phone near the heart [8]. While this is a topic of further research, it is better to avoid exposing subjects to mobile phone radiation. A wireless laptop-based phonocardiograph has been reported by Dao [9]. Building that type of consumer-grade device requires an investment of money and a high level of technological expertise.

With this background, the aim of this study was to build a low-cost, wireless phonocardiograph which can be used by keeping the mobile phone away from subjects.

## 2. Materials and Methods

This study was divided into two parts—(i) development of the wireless phonocardiograph, and (ii) recording of heart sounds in normal subjects.

### 2.1. Ethical Statement

The device was made by the first author with his personal household devices and instruments. In the second part of the study, heart sounds were recorded by a noninvasive method with negligible risk to the study subjects. The aim and procedure of the study were fully explained to the subjects in vernacular language before recruitment. After that, their written consent for participation was obtained. The recording of the female subjects was done in front of a female attendant with maintenance of full privacy and comfort. Recruitment of subjects and further study weredone according to full compliance with the Declaration of Helsinki 2013 after obtaining permission from the institute (No: 015/PHY/FMMCHB) [10].

### 2.2. Setting

This study was conducted in the Department of Physiology of a medical college in eastern India. The device was made during the month of September 2018 to October 2018. Heart sounds were recorded from 1–15 November 2018.

### 2.3. Materials Used for Development of the Modified Bluetooth Headset

Following are the details of the instruments used for development of the modified Bluetooth headset: (i) hacksaw blade (Carbon Steel, All hard, 18T, 300 mm) to cut the Bluetooth headset cover (i.e., cover), (ii) poly vinyl carbonate (PVC) electrical insulation tape (Anchor by Panasonic, Anchor Electricals Pvt. Ltd., Mumbai, India) to fix and fill the gaps on the device, (iii) instant adhesive (Fevi Kwik, Pidilite Industries Ltd., India) to fix the microphone with the circuit board of the device and to fix the bell with the microphone, (iv) stainless steel blade (Gillette Wilkinson Sword, Gillette (Shanghai) Limited, Shanghai, China) to cut the stethoscope ear knobs, (v) Bluetooth headset H904 (Chipset: ISSc2008, Bluetooth version: V4.1 + EDR, Supported profile: A2DP, AVRCP; Syska accessories, China) to record the heart sound, and (vi) soft rubber ear knob of a stethoscope (Elko Stethoscope Deluxe, Pace Technomed, India) to make the bell. The materials are shown in Figure 2.

### 2.4. Materials Used for Recording Heart Sounds

For recording heart sounds, the following materials were used: (i) newly modified Bluetooth headset, (ii) a mobile phone (Redmi note 4, 3GB RAM, 32GB internal storage, octa-core Max 2.0 GHz processor, Android version 7.0 NRD90M; Xiaomi, Beijing, China) to run the software which can record sound from Bluetooth headset, and (iii) Android application—Parrot version 2.4.12.184 (Searing Media Inc., Regina Canada) to record the sound on the mobile phone from the headset. The combination of the modified Bluetooth headset, mobile phone, and audio recording application can act as a wireless phonocardiograph (WiPCGh).

For visualization of sound wave patterns, a personal computer (PC) (Sony Vaio VGN-CS22GH; Sony, Tokyo, Japan) and a PC-based software application—Audacity^®^ (developed by a group of volunteers and distributed under the GNU General Public License) were used [11].

### 2.5. Procedure of Modification of Bluetooth Headset

The Bluetooth headset was modified by the following steps (Figure 3):(a)The cover on the opposite side of the speaker was opened.(b)The direction of the microphone was reversed. Normally the microphone is kept facing towards the face of the user when it is worn on the ear. We needed it to face the opposite direction.(c)The microphone was fixed to the spare part of the circuit board with the help of instant adhesive. This was done to make the microphone stable.(d)The ear knob of a stethoscope was cut circumferentially with the help of a blade on the side which normally faces the ear canal. This was done to make the opening wider.(e)The other end of the knob was put on the microphone. With a little push, the groove was fitted on the microphone. Then, the knob was also fixed to the circuit board with the help of instant adhesive.(f)The cover was measured according to the position of the knob on the circuit board.(g)The cover was cut with the help of the hacksaw blade.(h)The middle portion of the cover was discarded.(i)The rest of the cover was placed and fitted as it previously was. (j)Paper-based packaging materials were cut into pieces and used to fill the gaps between the ear knob and the cover.(k)PVC tape was cut into appropriate pieces and wrapped to fill the gaps.(l)The final modified Bluetooth headset device is shown with its intact earphone and the fitted bell.

### 2.6. Process of Recording Heart Sounds

Conveniently selected normal subjects (15 adult males and 15 adult females) were taken as a sample for the study. The Bluetooth device was switched on and connected to the mobile device. Then, the Parrot application was opened on the mobile. The mobile was kept in airplane mode to avoid any unwanted calls during the recording. After that, the rubber bell was wiped with surgical spirit and it was placed firmly on the precordium. To reduce noise, the position immediately adjacent to the apex beat was avoided. After securing a firm contact on the precordium, the recording was started on the Parrot app (44,100 Hz, WAV format, 5 dB gain, and skip silence off). After recording sounds for 1 min, the audio file was saved.

### 2.7. Processing of the Sound File

The sound files were shared from the mobile phone to a PC through Bluetooth connectivity. The sound files were opened in the Audacity^®^ software application to observe and analyze the wave form. Screenshots were captured by the Greenshot software application (Greenshot, a free and open source screenshot program for Microsoft Windows, Available from: http://getgreenshot.org/) to use in this manuscript.

## 3. Results

Figure 4a shows the modified Bluetooth headset with fitted bell, and Figure 4b shows a mobile phone with the Parrot application user interface. The modified Bluetooth headset has the following dimensions: length 5.8 cm, width 1.3 cm, depth 0.8 cm, bell height 1.1 cm, and bell diameter 1.3 cm at the widest part. These devices (Figure 4) were used to wirelessly record normal heart sounds in subjects of mean age 23.17 ± 4.05 years (male 23.27 ± 4.48 years, female 23.07 ± 3.73 years).

The wave pattern of the audio files obtained from the first two subjects (one male and one female) is shown in Figure 5. The wave between the first and second heart sound is the normal systolic and diastolic murmur.

## 4. Discussion

According to the aim of the study, a miniature, simple, and wirelessly controlled phonocardiograph was built. The cost for developing this device was minimal. This device can be built in resource-limited settings to record heart sounds for further use of the audio files. Simultaneously listening and visualizing the wave pattern would help medical students understand the first and second heart sounds. Hence, this WiPCGh may serve as an adjunct educational tool in teaching cardiovascular physiology.

The audio files (i.e., heart sounds) are recorded in the mobile device. They can be easily shared with anyone via a wide array of media (e.g., Bluetooth, email, WhatsApp). A study by Springer et al. found that mobile phone recorded heart sound is of sufficient quality for further analysis [12]. Hence, recorded audio files can be played or further analyzed on the receivers’ mobile phone or PC. This facility is also available for wire-connected devices [2,3,4,5]. In WiPCGh, as the Bluetooth headset is still a functional headset, the user can immediately play the audio file on the headset. This is an added advantage of WiPCGh. It is not possible on wire-based devices where the only microphone is used and the speaker of the earphone is sacrificed [2,3,4].

This miniature modified Bluetooth headset can be carried in pockets. As the bell is made up of an ear knob of a stethoscope, it can be wiped with an alcohol-based solution for cleaning purposes. When the bell is placed firmly on the chest wall, the skin acts as the diaphragm and the sound is conducted through the air column to the microphone. Thus, the sound is recorded. In previously developed devices [5], the same principle was used in a bell-based phonocardiograph. However, the stethoscope chestpiece-based phonocardiograph is equipped with a diaphragm (Figure 1).

The notable difference between the previously developed devices (Figure 1) and this device (Figure 4) is the wireless connectivity. This facility allows users to record the heart sound from a distance. This may help to avoid higher radiation emitted from the mobile phone [13]. For this study, when heart sounds were recorded, the subjects were instructed to keep the bell on the chest wall and the recording was done from a distance of 5 m. As the subject can hold the device with little instruction, it may ease the examination of female subjects with no touch and exposure to the recorder.

Comparative features of previously developed devices and this device are presented in Table 1.

In Figure 5b, the amplitude was higher than the male subject (Figure 5a). The age of the subject was similar (male 31 years, female 30 years). However, the chest wall thickness, relative position of the heart, and quality of sound may be the cause of this discordant finding. For the female subject, addition of +5 dB was more than required. Hence, this device may show different amplitude of waves for different individuals. To adjust it, the gain was reduced where the wave pattern was showing high amplitudes.

There are certain limitations of using this WiPCGh. This device does not have any noise cancelling properties as the commercially available stethoscopes have [14]. Hence, the quality of the recording may not be as impressive as what we can get from a high-cost digital stethoscope. However, selecting a calm and quiet room for recording may solve this issue. During this preliminary observation, heart sounds of normal subjects were recorded and the wave patterns were observed. Though this may serve as an exciting tool for educational purpose, its diagnostic capability is yet to be explored in future studies. High fidelity devices may identify third and fourth heart sounds [15]. However, WiPCGh was capable of recording the first and the second heart sounds. During development of the WiPCGh, a hacksaw blade and shaving blade were used for cutting the plastic cover of the Bluetooth headset and the stethoscope ear knob, respectively. These sharp instruments, if not handled properly, may cause injury. However, we did not experience any such events. In the current form of the WiPCGh, only a small bell was used to limit the size of the device. Attaching a Bluetooth headset with a stethoscope chestpiece may be a good approach.

## 5. Conclusions

A low-cost, wireless, and miniature phonocardiograph was made with easily available instruments and a Bluetooth headset. This device can be used to record heart sounds with minimal expertise. Medical teachers, science teachers, and doctors may record academically interesting heart sounds with this device on their mobile phone. Recorded audio files (i.e., heart sounds) can be easily shared via the web for the purpose of telemedicine. Further studies are required for testing the diagnostic utility of the WiPCGh.

## Figures and Tables

**Figure 1 medsci-06-00117-f001:**
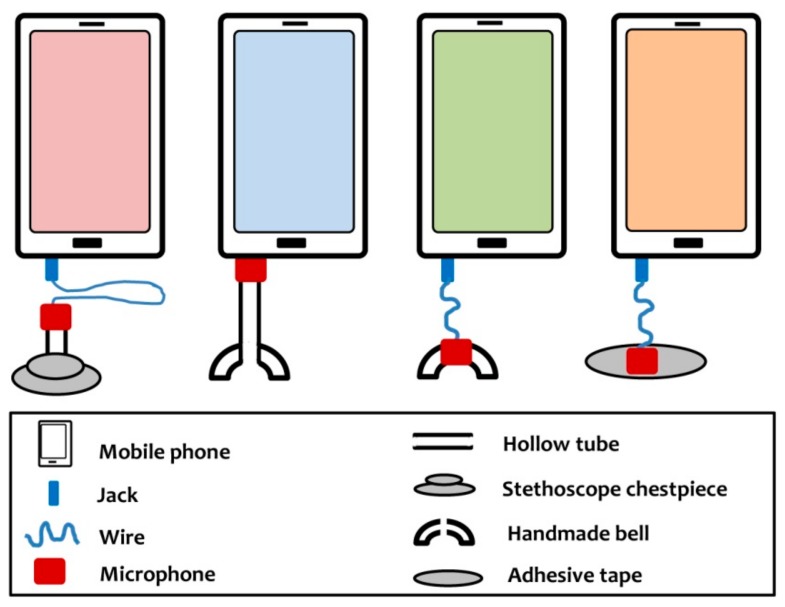
Types of low-cost, mobile phone-assisted, wired phonocardiograph.

**Figure 2 medsci-06-00117-f002:**
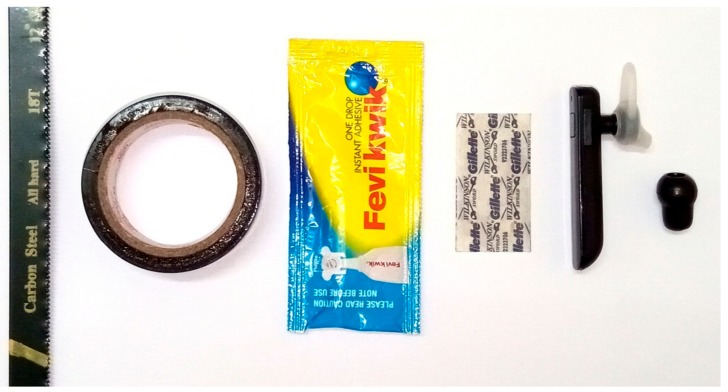
Materials used to transform a Bluetooth headset to a wireless phonocardiograph (From left to right: Hacksaw blade, poly vinyl carbonate (PVC) tape, instant adhesive, blade, Bluetooth headset, stethoscope ear knob).

**Figure 3 medsci-06-00117-f003:**
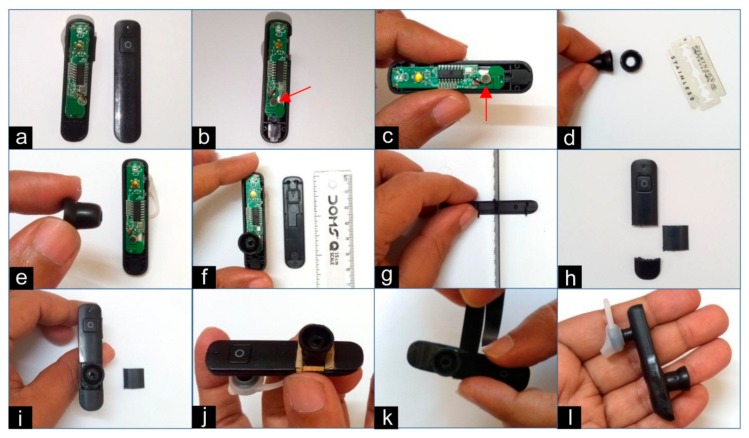
Steps of the modification of the Bluetooth headset: (**a**) opening the lid of the Bluetooth headset, (**b**) reversing the direction of the microphone, (**c**) fixing the microphone, (**d**) cutting a portion of the stethoscope ear knob, (**e**) placing the modified knob on the microphone, (**f**) fixing the knob, (**g**) cutting the headset lid, (**h**) discarding the unnecessary portion of the cover, (**i**) placing remaining parts to cover circuits, (**j**) putting in fillers, (**k**) fixing with PVC tape, and (**l**) the modified Bluetooth headset is shown.

**Figure 4 medsci-06-00117-f004:**
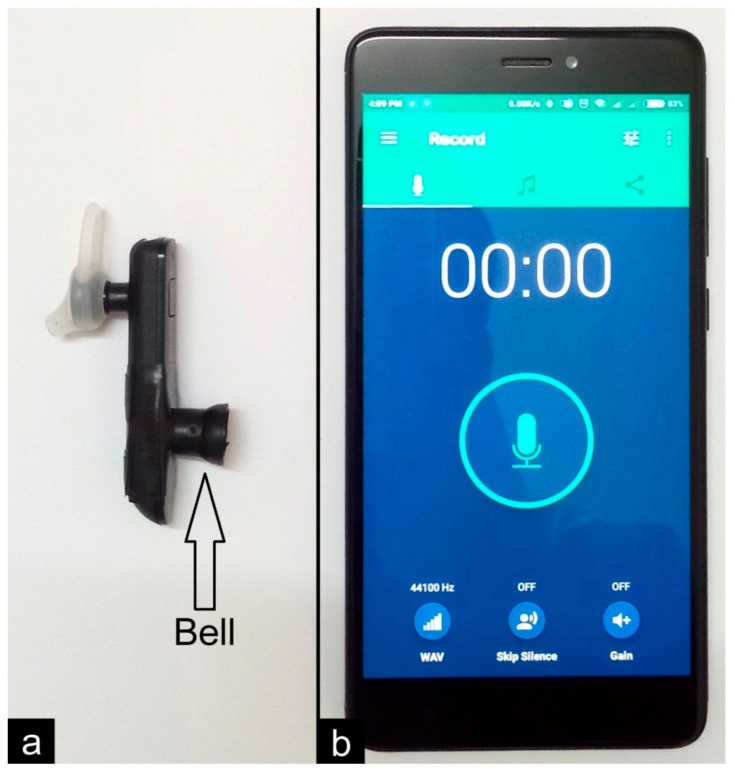
Developed wireless phonocardiograph (WiPCGh). (**a**) Modified Bluetooth headset; (**b**) mobile phone with audio recording software (Parrot user interface).

**Figure 5 medsci-06-00117-f005:**
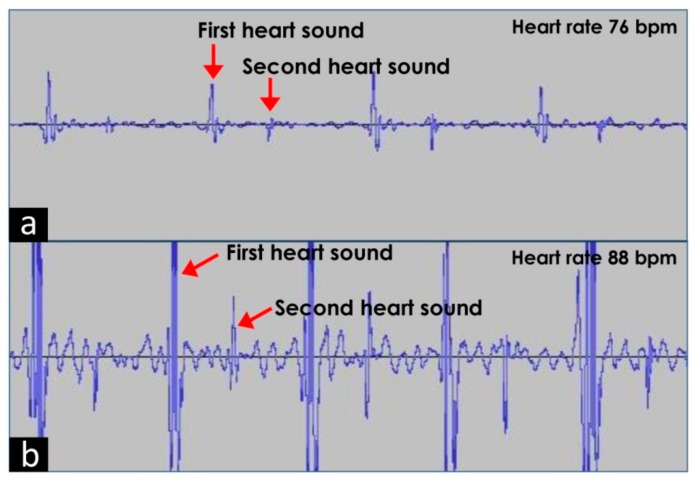
Waveform obtained from Audacity^®^ showing recorded heart sound of (**a**) an adult male and an (**b**) adult female. bpm: Beats per minute.

**Table 1 medsci-06-00117-t001:** Comparative list of different types of low-cost phonocardiographs.

Made/Reported by [Reference Number]	Type	Mobile Phone, Voice Recording Application and Instruments	How to Build
Finlay et al. [1]	Type I	Nil	Nothing to do
Bhaskar [2], Mamorita et al. [3], Bhimani et al. [4]	Type II	Stethoscope chestpiece, tube, earphone	Chestpiece is detached from the stethoscope tube. It is connected to a tube. The tube is connected to the microphone. The microphone is connected to a mobile phone via wire.
Thomas et al. [5]	Type III	Handmade bell, tube, microphone	The bell is connected to a tube. The tube is connected to the microphone. The microphone is connected to a mobile phone
	Type IV	Handmade bell, microphone	The bell is connected to the microphone. The microphone is attached to a mobile phone via wire
	Type V	Earphone, adhesive sticker	A hole is made on a sticker and the microphone of an earphone is attached to the sticker with the microphone facing the hole. The earphone is connected with a mobile phone via a wire
Current study	Type VI	Bluetooth headset, stethoscope ear knob as a bell	The microphone of a Bluetooth headset is attached to a bell. The headset is connected wirelessly with a mobile phone

Type is designated chronologically. Type I is not shown in any figure in this manuscript. Type II–V is shown in Figure 1 from left to right. Type VI is shown in Figure 4.

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
