# Peer review of "Development of a Low-Cost Wireless Phonocardiograph With a Bluetooth Headset under Resource-Limited Conditions"

_medsci, 2018, doi:10.3390/medsci6040117_

Round 1
Reviewer 1 Report
Summary
This manuscript describes the development of a low cost wireless phonocardiograph derived from a repurposed Bluetooth headset, audio recording smartphone app, and desktop audio processing software. The authors demonstrate use of the device to successfully capture the first 2 heart sounds, and suggest a higher fidelity device may be capable of detecting additional sounds.
General comments
1. The authors should be congratulated for their innovation and creativity. While use of this type of home-made device in diagnostic/treatment settings is unclear, the low cost and simple design of this method do hold potential in areas where more expensive equipment is not economically feasible and/or readily available.
2. One major concern is the lack of reporting about ethical approval. Was this research approved by a human ethics review committee? If not, the editors may need to consider the journal policy on publishing research without ethical approval.
3. The presentation of the manuscript could be improved with English language writing support. If they authors can access this type of support via their institutions I would encourage them to do so. In particular there are significant opportunities for more concise presentation of the content.
4. The authors suggest “wired” mobile phone-based PCG devices may be limited by close proximity of mobile phone microwave radiation to the heart but Bluetooth devices also use microwave radiation for data transmission, thus may be subject to similar concerns. Can the authors elaborate on this?
5. The manuscript could be strengthened by demonstrating not only that the device is can successfully record heart sounds, but also how it can be successfully used in education and/or clinical settings. Have the authors tried using it in the manner intended? Was the signal quality sufficient to use the device in the same was as a standard stethoscope? What benefits (if any) does this device offer that a standard stethoscope does not?
Specific comments
None.
Author Response
Comment: The authors should be congratulated for their innovation and creativity. While use of this type of home-made device in diagnostic/treatment settings is unclear, the low cost and simple design of this method do hold potential in areas where more expensive equipment is not economically feasible and/or readily available.
Reply: We thank the reviewer for her/his valuable comment about our manuscript. We agree with the reviewer that the diagnostic capability of this device is still not explored. However, the comment of the reviewer greatly influences us to do further study to test its diagnostic utility.
Comment: One major concern is the lack of reporting about ethical approval. Was this research approved by a human ethics review committee? If not, the editors may need to consider the journal policy on publishing research without ethical approval.
Reply: Madam/Sir, this study was conducted after getting ethical approval. We are sorry that we forgot to mention the approval number in the previous version of the manuscript. We have added it in the revised manuscript
Comment: The presentation of the manuscript could be improved with English language writing support. If they authors can access this type of support via their institutions I would encourage them to do so. In particular there are significant opportunities for more concise presentation of the content.
Reply: We agree with the reviewer that we have faced some problem in expression in English language. However, we are not capable to take language writing service due to lack of fund from us and from the institution. But we have tried our best to improve minor errors manually and detected by Ginger software. We request the reviewer to kindly consider our situation.
Comment: The authors suggest “wired” mobile phone-based PCG devices may be limited by close proximity of mobile phone microwave radiation to the heart but Bluetooth devices also use microwave radiation for data transmission, thus may be subject to similar concerns. Can the authors elaborate on this?
Reply: This is an interesting aspect pointed out by the reviewer. We fully agree with the reviewer that Bluetooth headset and mobile phone both use microwave radiation for signal transmission. However, the radiation of a Bluetooth device is very less when compared with a mobile phone. Due to long distance of the mobile tower, the radiation is high. After reading the comment of the reviewer, we thought that adding a sentence about it would enrich the article. We have also added reference number 13 for supporting the statement. The reference literature can be found at: https://www.oit.uci.edu/telephone/cell-safety/hands-free-devices/
Comment: The manuscript could be strengthened by demonstrating not only that the device is can successfully record heart sounds, but also how it can be successfully used in education and/or clinical settings. Have the authors tried using it in the manner intended? Was the signal quality sufficient to use the device in the same was as a standard stethoscope? What benefits (if any) does this device offer that a standard stethoscope does not?
Reply: We agree that the details about its use in teaching cardiovascular physiology could be added. This was briefly discussed in the first paragraph of the discussion. Though we made this device to show the wave patterns to students, we did not added details about the teaching procedure as we thought it might not be interesting as we did not use this in clinics. We request the reviewer to consider our limitation. If text about only physiological teaching is desired, we request the reviewer to provide us another chance to add text about it.
From a previous study it was found that mobile device recorded sound is of sufficient quality for further analysis (reference 12). However, we could not compare the quality and features of our device and digital stethoscope as we do not have that. Commenting about it in the manuscript would be unethical. We hope reviewer would consider our limitation about this

Reviewer 2 Report
In recent years, there have been a number of publications on using a smartphone to record the phonocardiography signals. However, until now, these studies made use of a wired microphone (or the microphone of the smartphone itself). The present manuscript describes a method where the microphone is connected wirelessly via bluetooth, which makes it much easier to employ and to deploy, especially in resource limited settings. Therefore the manuscript has its merits, although the idea of using bluetooth is pretty obvious.
- the method to modify the headset seems rather obtrusive and elaborative, and not easy for everyone to do.
- English can be improved
Author Response
Comment: In recent years, there have been a number of publications on using a smartphone to record the phonocardiography signals. However, until now, these studies made use of a wired microphone (or the microphone of the smartphone itself). The present manuscript describes a method where the microphone is connected wirelessly via bluetooth, which makes it much easier to employ and to deploy, especially in resource limited settings. Therefore the manuscript has its merits, although the idea of using bluetooth is pretty obvious.
Reply: We thank the reviewer for meticulously reviewing our manuscript and providing a positive comment about the manuscript.
Comment: The method to modify the headset seems rather obtrusive and elaborative, and not easy for everyone to do.
Reply:
Madam/Sir, we have tried to provide each and every minor details of the method so that anyone can follow the exact steps to make such device. As the manuscript is primarily based on the development of the device; hence, we used so much details. If it is decreasing the value of the manuscript, please allow us another round of revision. We will make the steps concise.
We agree that everyone cannot make it and there are chances of injury which we mentioned in the discussion. However, a little effort may give us a cheap but effective tool.
Comment: English can be improved
Reply: We have made corrections according to our personal capacity and also took help form a computer software for improvement of the manuscript.
